A candidate multimodal functional genetic network for thermal adaptation

Wollenberg Valero Katharina C. valerok@cookman.edu
Pathak Rachana
Prajapati Indira
Bankston Shannon
Thompson Aprylle
Usher Jaytriece
Isokpehi Raphael D.
College of Science, Engineering and Mathematics, Bethune-Cookman University , Daytona Beach, FL , USA
Amos William
Electronic publication date: 2014 Sep 30
Publication date: 2014
Volume: 2
Electronic Location ID: e578
Received 2014 May 2; Accepted 2014 Aug 24
Copyright: © 2014 Wollenberg Valero et al.
Copyright year: 2014
Copyright holder: Wollenberg Valero et al.
License: This is an open access article distributed under the terms of the Creative Commons Attribution License, which permits unrestricted use, distribution, reproduction and adaptation in any medium and for any purpose provided that it is properly attributed. For attribution, the original author(s), title, publication source (PeerJ) and either DOI or URL of the article must be cited.
License URL: https://creativecommons.org/licenses/by/4.0/

Keywords: Adaptation, Thermal adaptation, Reptiles, Thermoregulation, Ectotherms, Anolis, Genome mining, Functional network

Funding: NIH/NIMHD 1P20MD006899 NIH/NIGMS 5T36GM095335 NSF HBCU-UP 1435186 We received start-up funding from the Office of the Provost at Bethune-Cookman University/Faculty Initiatives to RDI and KCW. NIH provided funding to RDI (NIH/NIMHD 1P20MD006899; and NIH/NIGMS 5T36GM095335). NSF provided funding to KCW and RDI (NSF/HBCU-UP 1435186). The funders had no role in study design, data collection and analysis, decision to publish, or preparation of the manuscript.

==============================
Vertebrate ectotherms such as reptiles provide ideal organisms for the study of adaptation to environmental thermal change. Comparative genomic and exomic studies can recover markers that diverge between warm and cold adapted lineages, but the genes that are functionally related to thermal adaptation may be difficult to identify. We here used a bioinformatics genome-mining approach to predict and identify functions for suitable candidate markers for thermal adaptation in the chicken. We first established a framework of candidate functions for such markers, and then compiled the literature on genes known to adapt to the thermal environment in different lineages of vertebrates. We then identified them in the genomes of human, chicken, and the lizard Anolis carolinensis, and established a functional genetic interaction network in the chicken. Surprisingly, markers initially identified from diverse lineages of vertebrates such as human and fish were all in close functional relationship with each other and more associated than expected by chance. This indicates that the general genetic functional network for thermoregulation and/or thermal adaptation to the environment might be regulated via similar evolutionarily conserved pathways in different vertebrate lineages. We were able to identify seven functions that were statistically overrepresented in this network, corresponding to four of our originally predicted functions plus three unpredicted functions. We describe this network as multimodal: central regulator genes with the function of relaying thermal signal (1), affect genes with different cellular functions, namely (2) lipoprotein metabolism, (3) membrane channels, (4) stress response, (5) response to oxidative stress, (6) muscle contraction and relaxation, and (7) vasodilation, vasoconstriction and regulation of blood pressure. This network constitutes a novel resource for the study of thermal adaptation in the closely related nonavian reptiles and other vertebrate ectotherms.

Introduction

The human-induced climate change has recently fueled a renewed interest in the study of thermal adaptation (Angilletta, 2009; Angilletta, 2012). Temperature affects animals primarily via its effects on biochemical reaction rates (Hochachka & Somero, 2002). Tropical terrestrial vertebrate ectotherms such as lizards of the genus Anolis provide an excellent model group for the quantitative study of thermal adaptation to changing climatic conditions as we expect them to show an early, rapid response (Tewksbury, Huey & Deutsch, 2008; Deutsch et al., 2008; Huey et al., 2009). In order to understand the sum of effects of climate change on the fitness of organisms, and to create predictive models it is necessary to incorporate information on phenotype, physiology and evolutionary processes at the genomic level (Sears & Angilletta, 2011; Seebacher & Franklin, 2012). While behavioral and physiological responses to thermal changes are well studied, the exact molecular mechanism by which these responses are generated is still little understood. In the age of genomics, the next logical step is to integrate information from adaptive genomic markers with measures of physiological performance and selection, and to relate evolving genomic regions to physiological performance, in order to identify the pathways under which organismal responses to thermal changes are generated. Previous studies of adaptation have compared genomic divergence between populations adapted to differential environmental conditions (Hohenlohe et al., 2010; Hohenlohe et al., 2012; Hemmer-Hansen et al., 2013; Hübner et al., 2013). The challenge with this approach is that in addition to genomic adaptations to temperature differences, genomic regions of high divergence also can reflect divergence unrelated to this particular environmental variable, for example, differences in sensory environment (Gunter et al., 2011). To facilitate finding the connections between adaptive genomic regions and quantitative traits in relation to thermal adaptations, information on such adaptive pathways derived from the existing literature and online databases can be mined across evolutionary lineages, in order to identify possible candidate markers for adaptive responses to thermal environmental changes in non-model organisms. While candidate markers that correlate with thermal adaptation or plasticity could likewise be influenced by other underlying selective pressures, they provide a solid basis for the study of functional associations of markers diverging in population genomic studies. Based on literature mining of candidate markers related to thermal adaptation across evolutionary lineages, we construct functional genetic networks to determine functional associations between them. This will facilitate the discovery of genetic and cellular pathways relevant to thermal adaptation in ectothermic, vertebrate non-model organisms.

The organismal response to deviations from the thermal optimum can be subdivided in two categories based on their temporal dimension and potentially associated differences in the underlying molecular regulatory mechanisms. These are: (i) plasticity, which is the short-term response to thermal change and (ii) adaptation, which is a result of thermal change acting as an agent of natural selection on a population. Potential underlying mechanisms include hormonally regulated short-term responses, changes in regulatory pathways allowing permanent up-regulation of gene expression, and changes in sequences of genes and regulatory elements. In a current literature review, Urban, Richardson & Friedenfels (2013) found that plasticity played an important role in promoting phenotypic changes in response to short-term climate variation of ectothermic vertebrates (amphibians and reptiles). In that study, adaptive responses to short-term (human-mediated) climate change were not found, but climate adaptation occurred along spatial gradients, representing standing and historically more stable climatic clines. The influence of mechanisms allowing plastic responses of a population on adaptive evolution to more permanently altered thermal environments in amphibians and reptiles is still unclear (Urban, Richardson & Friedenfels, 2013). However, it is conceivable that evolutionary conserved molecular mechanisms and pathways can provide these short-term changes of body physiology. Especially when combined with standing variation in a population, this might be considered important raw material for adaptation to occur (West-Eberhardt, 2003). For example, plasticity has been predicted to allow birds to adapt to climate change (Vedder, Bouwhuis & Sheldon, 2013). Known molecular genetic markers that have been identified as having a function in plastic response or adaptation in any clade therefore serve as an ideal basis to identify functional genetic pathways of thermal adaptation.

Mammals and birds are most commonly referred to as endotherms, as they are able to generate body heat from metabolic processes, with a set-point optimal temperature (homeothermy). Other vertebrate lineages, fishes, amphibians and reptiles, are referred to as ectotherms since they usually do not maintain a stable body temperature (heterothermy). However, exceptions to this are numerous: heterothermy in the form of torpor (hibernation) is present in many endotherms (e.g., Madagascan Lemurs, Dausmann et al., 2004; birds, McKechnie & Lovegrove, 2002).

Endotherm mammals generate heat (nonshivering thermogenesis) by lipolysis (breaking up fat in the adipose tissue). During cold exposure, the neurotransmitter noradrenaline is released to brown adipose tissue (BAT) where lipolysis takes place (Leppäluoto et al., 2005). Free fatty acids open a mitochondrial proton channel (Thermogenin, coded by the UCP1 gene), which results in protons returning to the intermembrane space and inhibiting ATP synthesis (uncoupling). Instead, energy is lost as heat (facultative or adaptive thermogenesis, reviewed in Leppäluoto et al., 2005). BAT is the thermogenic component of the sympathethic nerve system. In ectotherms, it is not known which mechanisms are adaptive to changes in thermal environment except for behavioral thermoregulation. BAT is absent in marsupials and monotremes (Hayward & Lisson, 1992). Heat generation in endotherms is related to lipid transport and often expressed in BAT. In ectotherms, BAT is not present, and the role of thermoregulation as associated with the lipoprotein metabolism is not clear. Endotherm birds also use the uncoupling mechanism but since BAT is absent in them, the primary location for nonshivering thermogenesis is the skeletal muscle (reviewed in Leppäluoto et al., 2005).

For ectotherms, the predominant view is that mostly morphological or behavioral adaptations exist to enable heat or cold plasticity and adaptation. These include convection (heat loss by increasing blood flow close to the body surface, conserving heat by decreasing blood flow close to the body surface, Bartholomew & Tucker, 1963), conduction (moving close to a surface with warmer/colder temperature), radiation (minimizing/maximizing sun exposure), and insulation (altering the surface/volume ratio, e.g., by plumage). Behavioral thermoregulation is undoubtedly the main mechanism to buffer changes in thermal environment (Sunday et al., 2014). However, present species distributions of ectotherms sometimes span large ranges in thermal environments, and local populations do show adaptations to these local thermal conditions. Genetic mechanisms must therefore exist that explain such local adaptations. For example, many amphibians and nonavian reptiles (in the following: reptiles) are found in diverse thermal environments (Addo-Bediako, Chown & Gaston, 2000), and several studies report examples of adaptation to such different thermal environments (e.g., Muñoz et al., 2014). The purpose of this paper is to elucidate candidate genes and functional pathways in which thermal adaptation takes place beyond behavioral thermoregulation in vertebrate ectotherms.

Heterothermy has been proposed to be plesiomorphic originating in the common ancestor of birds and mammals (putatively a therapsid reptile). If this is the case, it would mean that the same pathways for thermal plasticity and adaptation that are known from birds and mammals are likely to also operate in extant reptiles (reviewed in Grigg, Beard & Augee, 2004). Other studies found that ectotherms seem to be able to regulate their internal temperature set-points at least to a small degree. For example, Anolis lizards are able to acclimate their critical thermal minimum in the course of only a few days (CTmin, Petersen, Gleeson & Scholnick, 2003; Kolbe et al., 2014), as do alligators (Guderley & Seebacher, 2011). Other adaptations to diverse thermal environments include for example the presence of cryoprotectants in the blood stream (such as glycerol, in fishes, frogs, Zimmerman et al., 2007; Honer et al., 2013). Thermal plasticity and adaptation in reptiles might thus be mediated via the same genes and cellular pathways that respond to thermal environmental changes in other lineages such as boid snakes, birds and mammals. Based on information presented in the literature reviewed in this section, we expect candidate markers for thermal adaptation (and, possibly, plasticity) in ectotherms such as Anolis lizards, to fall into one of five functional categories:

1. Associated with the lipoprotein metabolism/nonshivering thermogenesis.

2. Associated with membrane channels controlling water loss/retention or cryoprotectants.

3. Associated with short-term stress response.

4. Associated with phenotypic or phenological changes arising as a consequence to thermal change, e.g., pigmentation associated with thermoregulation.

5. Associated with thermal signal transduction.

In this paper, we test the hypothesis that previously identified candidate genes for thermal adaptation can be grouped into these proposed functional categories and consequently are functionally connected via gene interaction pathways. To test this hypothesis, we integrate literature searches of known markers for thermal adaptation with functional association modelling in order to identify genes that have found to be under selection (adaptation), or are known to show a short-term response (plasticity) to changes in the thermal environment. Candidate genes are retrieved from studies performed in any vertebrate including humans, lizards, chicken, frog and fish. We identify functional interactions between them using the chicken genome as a model. A flowchart of the workflow of this paper is presented in Fig. 1. Since the chicken is the model organism closest related to nonavian reptiles, we discuss the possible relevance of candidate genes for them (especially Anolis lizards, Fig. 2).

Materials & Methods

Genes that are known to be related to thermal plasticity or adaptation in any vertebrate lineage were identified via literature searches in PubMed (http://www.ncbi.nlm.nih.gov/pubmed), GeneCards (http://www.genecards.org/), and ENSEMBL (http://www.ensembl.org/index.html) in March 2014, and matched to gene identifiers of the Homo sapiens (human), Gallus gallus (chicken), and Anolis carolinensis (lizard) genomes. The search terms for finding candidate genes were “thermal adaptation/tolerance”, “heat adaptation/tolerance”, “cold adaptation/tolerance”, and “climate adaptation”. To avoid circularity of our argument, no gene function was used as a search term. Table S1 shows a list of the identified candidate markers with their location on the Anolis carolinensis genome, and a list of functions of the human orthologs of these candidate genes were retrieved from RefSeq (http://www.ncbi.nlm.nih.gov/refseq/) via GeneCards. Functional relationships that are known or predicted between encoded proteins of the potential candidate genes were determined via the STRING algorithm (V.9.1) embedded in CYTOSCAPE (V.3.1.0, Shannon et al., 2003) for human and chicken. CYTOSCAPE visualizes functional interactions of genes by applying a network-based algorithm based on molecular triangulation (Krauthammer et al., 2004). Functional pathways between genes were inferred by (1) interologs (= protein interaction across evolutionary lineages) mapping, (2) curator inference, (3) predictive text mining, (4) phylogenetic profile, (5) experimental interaction detection. Not all functional genetic interactions that are known for humans, are known for the chicken yet. Four additional functional connections that connect parts of the candidate networks in humans but not in the chicken were therefore manually added to the algorithm-generated chicken networks in CorelDraw (V.X6). It is important to point out that these four connections are hypothetical until further evidence is available.

In order to test whether our predicted gene functions are gene functions which are more common than expected by random functional interactions (called statistically overrepresented Gene Ontologies, GO), we applied the Hypergeometric test using the BiNGO plugin to CYTOSCAPE (Maere, Heymans & Kuiper, 2005). BiNGO maps the predominant functional themes of a given gene set on the GO hierarchy and applies the Hypergeometric test (Maere, Heymans & Kuiper, 2005). The Benjamini & Hochberg (1995) False Discovery Rate (FDR) correction was applied for multiple tests.

To test the hypothesis that candidate genes are indeed more strongly associated with candidate functions than associated genes not initially postulated as candidates, a randomization approach was used. A list of 395 genes that were the closest neighbors (protein-coding genes with known function) of candidate genes in the candidate gene functional network was established as a new dataset. From this list of neighbor genes functionally associated with candidate genes but not initially postulated as candidate genes, we extracted 50 random subsets of 44 genes each (corresponding to the initial number of candidate genes), and calculated functional networks from them. Network statistics were then computed with the NetworkAnalyzer function in CYTOSCAPE from both the candidate gene network and the random networks. Ozgur et al. (2008) found that such centrality measures can significantly predict disease association for candidate genes in a network. We tested for difference in the metrics clustering coefficients, heterogeneity, network density, and average node closeness centrality between randomized networks and candidate gene network.

To test whether candidate functions are more present in the candidate network than in the random networks, BiNGO was used to calculate significantly overrepresented Gene Ontologies from the random networks. A Mann–Whitney U Test (with continuity correction) was performed in STATISTICA (StatSoft, Tulsa, OK) to test for differences in presence/absence of candidate functions of genes and gene interaction groups across the two comparison groups (random neighbors and candidate genes).

An interactive web-based visual analytics resource was developed to facilitate data exploration of the gene list for knowledge-building on thermal adaptation in vertebrates. The web-resource is available at: https://public.tableausoftware.com/views/thermal_adapt/gene_list.

Results and Discussion

31 of our initial list of 44 candidate markers were retrieved from the chicken genome. All these markers were functionally related, by association either through the CYTOSCAPE software, or by manual association after comparisons of functional pathways with the human equivalent of the functional network. The chicken network is shown in Figs. 3–5. A more detailed version of the large functional network (Fig. 3) is shown in Fig. S1.

Figure 1 Flow chart of the process of identifying candidate functional pathways that are likely adaptive to changes in the thermal environment across vertebrates. Question marks denote hypothesis testing.

Candidate genes for thermal adaptation: Genes that have been found to be adaptive (either by sequence modification or by changes in expression levels) related to changes in the thermal environment. Functional Genetic Pathways: All DNA segments in an organism that directly or indirectly interact with each other to perform a cellular function. Genetically Overrepresented (GO) functions: Abbreviation for Genetically Overrepresented Gene Ontology, which describes which functions are common in a genetic regulatory network. This includes the functions performed by single genes, as well as functions performed by interacting genes. Vertebrate Ectotherms: Fish, Amphibians, and Reptiles. Yellow boxes denote research resources provided in this paper.

Figure 2 Amniote phylogeny based on 3,994 one-to-one orthologous synonymous protein sites showing major features of amniote evolution.

Printed with permission from Alföldi et al., Nature 2011.

Figure 3 Large functional network of a subset of candidate marker genes for thermal adaptation.

Candidate markers are indicated in green, links to other networks (depicted in Figs. 4 and 5) in pink. Hypothetical connections manually inferred from human genes to connect parts of chicken networks (NPS) are shown in the hexagonal box. A detailed version of this figure can be accessed in Fig. S1.

Figure 4 Aquaporin and Heat shock protein gene functional networks of a subset of candidate marker genes for thermal adaptation.

Candidate markers are indicated in green. Hypothetical connections manually inferred from human genes to connect parts of chicken networks (NPPA, SUMO1) are shown in hexagonal boxes. Dashed lines—hypothetical functional association with the large network depicted in Fig. 3.

To test whether candidates are in closer functional association with each other than similar genes not proposed as candidates, we compared clustering coefficients, heterogeneity, network density, and average node closeness centrality between randomized networks and candidate gene network (Fig. 6). The network clustering coefficient is a measure of how well nearest neighbors of candidate genes are connected. The candidate network was significantly less clustered than the randomized neighbor networks, hinting at discrete functional pathways instead of genes shared across different adaptive functions in the candidate gene network. The network heterogeneity describes the tendency of a network to contain hub nodes (Dong & Horvath, 2007). Despite identifying genes that function in thermal signal relay and stress response, this measure was not significantly different from the randomized neighbor networks. Network density describes the propensity of network nodes to be isolated vs. forming a clique (being more functionally associated). The candidate gene network was significantly denser than the random neighbor networks, indicating closer functional association between candidate genes. Closeness centrality is a measure of how fast information spreads from a given node to other reachable nodes in the network (Newman, 2005). The closeness centrality averaged over all nodes was significantly higher in the candidate gene network, showing that functional associations among candidate genes were higher than expected by chance in functionally associated non-candidate genes.

Figure 5 Functional networks of the ADORA Functional Network.

Candidate markers in green. Dashed line—functional association with the large network depicted in Fig. 3.

Figure 6 Comparison of (A) clustering coefficients, (B) network heterogeneity, (C) network density and (D) average closeness centrality of candidate and 50 randomized networks.

Candidate gene network neighbors (bar) are significantly less connected, nodes are equally heterogeneous, the network is significantly more dense (= functionally related), and has a significantly larger closeness centrality than the randomized neighbor networks (columns).

23 of the 31 markers in the candidate gene network (Figs. 3–5) were represented as having a significantly overrepresented GO (Table 1). Among our five predicted functions for these candidate markers, four were retrieved as overrepresented GOs (Table 1). The only predicted function of candidate markers that was not retrieved as overrepresented GO was that of phenotypic change associated with thermal change, as in the example of pigmentation associated with thermoregulation in reptiles. The frequently studied MC1R gene that is associated with such pigmentation and that was in our list of potential candidate markers retrieved from the literature, could not be retrieved in the chicken network, nor could it be located in the Anolis carolinensis genome. Instead, the MC2R-4R genes were represented in the network, but not retrieved as a significantly overrepresented GO and are discussed below. Another group of genes that formed part of the candidate gene list and functional network but were not overrepresented GOs is the AQP gene family coding for water and glycerol channels. One possible reason for this is that maybe their relative number in the network is too small, or the functions in the chicken are still understudied or remain to be annotated, and we here flag them for further study. Three additional statistically overrepresented GO functions retrieved by BiNGO were not among our functional predictions. These additional, previously unpredicted functions related to thermal plasticity and adaptation are (Table 1):

6. Response to oxidative stress

7. Muscle contraction and relaxation, and muscle development

8. Vasodilation, blood circulation and blood pressure regulation.

Table 1 Predicted and retrieved genetically overrepresented functions of candidate markers for thermal adaptation in a functional genetic network constructed for the chicken.

Error probabilities for genetic overrepresentation are derived from the test for Hypergeometric distribution, after Benjamini–Hochberg correction for multiple samples.

Predicted functions	Retrieved significantly overrepresented
Gene Ontologies (p < 0.05)	
Associated with the lipoprotein metabolism	LPL, CD36, CETP, MAPK1, MAPK14, SOD1, STUB1, LEPR, UCP3	
Associated with membrane channels controlling water loss/
retention or cryoprotectant (glycerol).	LPL, CETP	
Associated with stress response	MAPK1, UCP3, HSF1, MAPK14, HSP47, UNG, HSPB2, SOD1,
STUB1, HSPA8	
Associated with phenotypic or phenological changes arising
as a consequence to thermal change, e.g., pigmentation	Not overrepresented	
Associated with signal relay	MAPK1, UCP3, HSF1, ADORA2B, ADRB2, MAPK14, HSP47,
UNG, HSPB2, MC4R, SOD1, STUB1, HSP48, EGFR, CD36,
MRAS, ADORA1	
Additional functions, not predicted		
Associated with oxidative stress response	UCP3, SOD1	
Muscle contraction and relaxation, muscle development	ADORA2B, SOD1, HSPB2	
Vasodilation, blood circulation, blood pressure regulation	ADORA2B, SOD1, POMC	

The genes providing a response to oxidative stress might be associated with the fact that low temperatures cause hypoxia (Petersen, Gleeson & Scholnick, 2003), and both high and low temperatures cause oxidative stress (free oxygen radicals in the cell). Freeze-tolerant reptiles display tolerance for hypoxia and antioxidant defense (Storey, 2006). For example, freeze responsive genes in turtles code for (a) proteins involved in iron binding, (b) enzymes of antioxidant defense, and (c) serine protease inhibitors, that all are functionally related to providing oxygen and glucose to tissues under hypoxia (Storey, 2006).

The response to thermal changes associated with genes involved in muscle contraction and relaxation clearly points at the connection between nonshivering thermogenesis in birds, and ectothermic shivering thermogenesis, both processes being located in the muscle. Shivering thermogenesis is the only facultative way of generating body heat and is universally found in mammals and birds, as well as some reptiles (boid snakes). In shivering thermogenesis, body heat is generated in the skeletal muscles by neuronally controlled muscle contractions (Hutchison, Dowling & Vinegar, 1966; reviewed in Grigg, Beard & Augee, 2004). This process can elevate body temperature in brooding female pythons up to six degrees Celsius (Brashaers & DeNardo, 2013) and proves that endo-and ectothermy are not perfectly delimited categories.

Vasodilation, vasoconstriction, blood circulation and blood pressure regulation have been observed to be important mechanisms in reptiles related to convection. Heat loss in reptiles is accomplished by increasing blood flow close to the body surface, versus conserving heat by decreasing blood flow close to the body surface (Bartholomew & Tucker, 1963). Genes associated with vasodilation and -constriction can regulate such a mechanism. Although vasoconstriction was not among the significantly overrepresented GO, one of the candidate genes (EDN1) is functionally associated with vasoconstriction in humans.

To test whether significantly overrepresented GO that correspond to candidate functions were more present in the candidate gene network than in the random neighbor networks, we tested for presence versus absence of candidate functions in both test groups. After removal of redundant functions per gene/interaction group (remaining N = 3,048), the candidate gene network recovered significantly more genetically overrepresented candidate functions than the random networks. This results corroborates the finding that the association between candidate genes and postulated functions is significantly higher than expected by chance in functionally associated non-candidate genes (Table 2). In the following sections candidate genes that are represented in our functional genetic network and potentially relevant for genetic adaptations to thermal changes are discussed. Functions that are not cited in a reference correspond to functions of the human gene ortholog/s obtained from RefSeq and deposited in Table S1.

Table 2 Mann–Whitney U test (with continuity correction) for difference in presence/absence of candidate functions retrieved from candidate gene network and randomized networks.

The candidate gene network recovered significantly more GO candidate functions than the random networks.

U	Z	p-value	Valid N randomized
neighbors	Valid N candidate genes	
146,764.5	−8.285	1.187∗E−16	2,883	165	

Candidate markers within the lipoprotein-metabolism associated functional network

The largest functional network that was recovered from the chicken genome, falls into several functional categories (Fig. 3). Several of these candidate markers have been identified in a study of SNPs in the global human population with respect to their variation with global climate (Hancock et al., 2008). The study recovered several SNPs within genes for common metabolic disorders that were associated with latitudinal variation (FABP2), summer duration (CD36, DSCR1, MAPK14, PON1, SOD1, CETP, EGFR, and NPPA) and winter duration (RAPTOR, UCP3, LPA, MMRN1, EPHX2, LEPR, MAPK1, Hancock et al., 2008). However, the Hancock et al. (2008) study did not include network-based visualization, so that the functional relationships of these markers to each other were not recovered. We here discuss these and other markers based on their functional associations within the chicken candidate network. The numbers of sub-headings correspond to the network parts described in Fig. 3, and to the function of that part of the network (denoted “GO” if significantly overrepresented).

1—Signal relay (GO), stress response (GO), Lipoprotein metabolism (GO). MAPK1 and MAPK14 are genes that are important for several cellular signaling pathways, and their sequence variation is related to winter adaptation by humans (Hancock et al., 2008). Human MAPK14 furthermore is specifically activated in response to environmental stresses.

2—Signal relay (GO). The EGFR protein is a membrane receptor for epidermal growth factors. It is functionally related to seven other candidate markers in our network, and could therefore represent a functional hub for the relay of thermal signaling. In humans, it is associated with lung cancer which relates it to respiration.

3—Lipoprotein metabolism. The LEPR gene product (the leptin receptor), has been shown to significantly vary with the duration of winter in the human population, which represents an adaptation to cold environments (Hancock et al., 2008). According to Hancock et al. (2008), LEPR is a strong candidate gene due to its involvement in a thermogenesis pathway that is inducible in skeletal muscle in the mouse model (Dulloo et al., 2002; Dulloo, Seydoux & Jacquet, 2004; Hancock et al., 2008)), bearing similarities to nonshivering muscle-associated thermogenesis in birds and shivering muscle associated thermogenesis in birds and reptiles. LEPR is associated with EGFR, MAPK1 and, via POMC, with five other markers functionally associated with thermogenesis.

4—Vasodilation (GO), blood circulation (GO), blood pressure regulation (GO). POMC, the gene for the pro-opiomelanocortin receptor, undergoes tissue specific posttranslational processing. The functions of the resulting peptides in humans include maintenance of adrenal weight, inflammatory pain and energy homeostasis, melanocyte stimulation, and immune modulation. In carp fish (Cyprinus carpio) that is a eurytherm species persisting over a wide range of temperatures, two POMC genes are present. Their expression levels have been found to alternate with temperature (24 versus 9 °C; Arends et al., 1998). An interesting experiment performed by Chuang et al. (2004) showed that human POMC gene introduced via plasmids into muscles of arthritic mice alleviated both thermal hypersensitivity and paw swelling symptoms. Swelling hints at its involvement in vasodilation, which was a significantly overrepresented GO for this gene. POMC mutations in humans are associated with early onset obesity and linked to leptin concentrations (Delplanque et al., 2000). This also corroborates the role of POMC in thermal signal relay and its putative functional association with the lipoprotein metabolism (LEPR gene). Another candidate gene that was among our search terms was MC1R (the melanocortin 1 receptor). The interaction of MC1R and POMC expression (a hormone that can stimulate the MC1R) is known, with available amplification primers available for several species of reptiles and amphibians (e.g., Ducrest, Keller & Roulin, 2008). POMC signals to the melanocortin receptor lead to the movement or permanent positioning of melanin through layers of the skin, causing changes in pigmentation as well as pattern phenotypes. Since skin darkening is a major component of thermoregulation in reptiles (increasing the absorbed sun radiation), this gene complex is an ideal candidate gene pair to study thermal plasticity and adaptation (Rosenblum, Hoekstra & Nachman, 2004). However, this gene was not retrieved in the chicken network (despite supposedly being present on chromosome 11, Table S1), and also was not found in the Anolis carolinensis genome. Instead, among the genes directly interacting with POMC in the chicken functional network were MC2R, MC4R, and MC5R. MC5R could not be found in the lizard genome, but MC2R and MC4R have orthologs in the lizard genome, and were therefore post-hoc included into our list of candidate genes (Table S1). MMRN1 is another candidate marker with direct functional link to POMC that has been shown to vary with winter duration in the global human population (Hancock et al., 2008). It encodes for a large soluble protein, multimerin, found in platelets or the endothelium of blood vessels. It may play a role in cell adhesion. A paralog, MMRN2, was identified as providing adaptation to high altitude in Yak (Bos mutus, Qiu et al., 2013). It functionally relates to AHSG, as well as to the ADORA gene network.

5—Lipoprotein metabolism (GO), stress response (GO), signal relay. STUB1 is involved in the degradation of misfolded proteins, and can modulate the activity of several heat shock proteins. It is involved in the cellular reaction cascade that responds to heat stress. Its involvement in thermal plasticity and adaptation in non-mammalian vertebrates has not been studied yet.

6—Vasoconstriction, oxidative stress response. EDN1 encodes the endothelin-1 peptide in humans which is important in vasoconstriction. It is involved in hypoxic pulmonary vasoconstriction where pulmonary arteries constrict in the presence of hypoxia (low oxygen levels which leads to redistribution of blood flow to better-ventilated areas of the lung, which increases the total area involved in gaseous exchange. It has been demonstrated to be involved in high altitude adaptation (Savourey et al., 1998), for example in vascular adaptation to high altitudes in pregnancies (Moore et al., 2004). Length variants in endothelin-1 have been shown to be associated with altitudinal acclimation as well (Rajput et al., 2006). The EGLN1 marker is another candidate gene related to the functional association with oxygen transport. In humans, adaptations in this marker have been found in populations that inhibit high altitudes, i.e., experience reduced oxygen levels (Aggarwal et al., 2010; Peng et al., 2011; Xiang et al., 2013; Mishra et al., 2013). EGLN1 is related to EPAS2, which in humans encodes a transcription factor involved in the induction of genes regulated by oxygen, which is induced as oxygen levels fall.

7—Signal relay (GO) MRAS (also called R-Ras3) plays an unknown physiological role in humans (Labunskay & Meiri, 2006). In our functional network it was placed as relaying the signal from another gene, EGFR, and from MAPK1 via the JUN gene which is also part of our predicted thermal signal relay cascade. A previous study has shown that it is involved in the establishment of thermal control in the chicken embryonal brain. Both heat and cold induce gene expression of MRAS and of JUN (Labunskay & Meiri, 2006), corroborating its putative function as a thermal signal relay gene.

8—Stress response (GO), oxidative stress response (GO), Signal relay (GO), Muscle contraction and relaxation (GO), vasodilation (GO). SOD1 codes for superoxide dismutase, an enzyme that can reduce the concentration of free superoxides in the cytoplasm. Oxidative stress is thought to increase with deviation from the thermal optimum in fish (Vinagre et al., 2012). A study in shrimp has shown that the shrimp equivalent of the SOD1 gene is upregulated under high heat stress, putatively via heat shock proteins (Sookruksawonga, Pongsomboona & Tassanakajon, 2013). We could show in this contribution that SOD1 is indeed functionally related to the HSP gene family (see HSP discussion below). SOD1 being involved in short-term cell protection in response to thermal stress, but being controlled by the same regulatory pathway (MAPK1 via STUB1) which relays the signal for thermoregulation and is adaptive to thermal environmental changes, indicates that there is a functional relationship between plasticity and adaptation, and furthermore to its evolutionary conserved function as proven by functional similarity in shrimp, fish and humans.

9—Stress response (GO). UNG codes for two alternatively spliced DNA Uracil glycosylases that are involved in mismatch repair of Uracil in DNA. The UNG gene is functionally related to SOD1 which itself is influenced by the Chaperone STUD1. Previous studies have found UNG to be cold-adapted in Atlantic cod fish with an increased catalytic efficiency and thermoliability as compared to UNG adapted to medium temperatures (Lanes et al., 2000; Olufsen, Smalås & Brandsdal, 2008; Assefa et al., 2012).

10—Stress response, response to oxidative stress, Lipoprotein metabolism. AHSG is functionally related to MMRN1 and PLG. AHSG is expressed in the human liver, secreting the AHSG glycoprotein. It is also expressed in adipose tissue in humans, and is involved in the predisposition to obesity (Dahlman & Arner, 2007). It is a part of the lipoprotein metabolism, and interesting as a candidate gene due to its functional association with other markers. The HPSE gene codes for heparanase which is hypothetically related to AHSG via the human NPS gene. Heparanase is involved in the constitution of the extracellular matrix and cell–cell interactions. Adaptive evolution of heparanase has led to a unique splice variant in the rodent Spalax ehrenbergi, which is presumably involved in adaptation to hypoxia, or metabolic stress in general (Nasser et al., 2005).

11–13—Lipoprotein metabolism (GO), water channel/glycerol as cryoprotectant (GO), signal relay (GO). CETP is a marker positively associated with adaptation to summer duration in humans (Hancock et al., 2008). Single Nucleotide polymorphism frequencies within the human FABP2 gene, encoding a protein that binds fatty acids, have been shown to be significantly varying with latitude of the global human population (Hancock et al., 2008). FABP2 variants are adaptive to thermal changes in terms of increasing fat storage in cold environments (Hancock et al., 2008). LPL (lipoprotein lipase) is an enzyme involved in the breakdown and uptake of lipoprotein triglycerides related to the lipoprotein metabolism which in the mammal BAT is a main pathway for nonshivering thermogenesis. Jensen et al. (2008) found that transgenic mice who overexpressed human LPL in their skeletal muscle displayed enhanced cold tolerance and thermogenesis by increased fat oxidation. Jensen et al. (2008) suggested that this response was achieved by gene expression in skeletal muscle. This phenotype resembles that of birds which exhibit a thermogenic response to cold temperatures via their skeletal muscles. Consequently, the LPL gene is a good candidate to study non-BAT associated thermogenesis in birds, and potentially in vertebrate ectotherms. CD36 varies significantly in humans with the intensity of summer (Hancock et al., 2008). It encodes for the thrombospondin receptor, binds to oxidised LPL and might function as a regulator or transporter of fatty acids. Its intermediate position in the FABP2 → LPL → CD36 → UCP functional connection points at its involvement in BAT-associated thermogenesis. The Uncoupling Protein 3 gene is the fourth component of the functional network constructed with the known candidate markers for thermal adaptation, the UCP gene. Numerous studies have identified uncoupling proteins (also called mitochondrial anion carrier proteins) as central players in mammalian BAT-located thermogenesis. In endotherms, the BAT-expressed UCP is UCP1. UCP3 has tissue-specific transcription initiation upstream of the SM-1 (major skeletal muscle site, Esterbauer et al. (2000). This associates it with skeletal muscle-associated nonshivering thermogenesis, as it has been found in previous studies from both mammals and birds (e.g., Klingenspor, 2003). For example, Teulier et al. (2010) found that ducklings reared under lower ambient temperature were cold-acclimated by upregulation of UCP, and had a higher capacity for muscle associated nonshivering thermogenesis. King penguins have been shown to respond to cold stimuli with thermogenesis both by uncoupling oxidative phosphorylation in the mitochondria of the skeletal muscle by expressing UCP and increased proton transport activity of the adenine nucleotide translocase (Talbot et al., 2004). In this regard the candidate gene mt-ATP6 coding for mitochondrial ATP-Synthase might be of interest—but this marker was not connected with the general interaction network of the chicken. Besides plasticity, UCP has been found to have undergone adaptive evolution: UCP orthologs are present in all vertebrate lineages including fishes, and UCP1 has undergone rapid diversification in ancestral eutherian mammals (Saito, Saito & Shingai, 2004) associated with BAT-associated thermogenesis. However, the gene duplication events leading to the three paralogs UCP1, UCP2, and UCP3 were older than the diversification of eutherians (Saito, Saito & Shingai, 2004), as confirmed here by the presence of UCP3 in the chicken functional network.

Candidate markers within the heat shock protein gene functional network

Stress response (GO), signal relay (GO), Muscle contraction and relaxation (GO). Heat shock proteins (HSPs) are molecular chaperones that are universally present in all organisms and carry out a well characterized stress response: HSPs protect proteins that denature under heat stress from aggregating (Tavaria et al., 1996). The Heat shock protein network could manually be linked to the large functional network via HSPA1 and HSPA4 being functionally connected to MRAS and SOD1. Narum et al. (2013) investigated the heat shock response in montane and temperate strain of fish and found high induction of heat shock proteins in the montane strain, which appeared to improve short-term survival during first exposure to high water temperatures. However, this was not associated with an increased long-term survival of fish under thermal stress which underlines the function of the HSP response as being plastic, not adaptive. In contrast, Garbuz et al. (2008) studied HSP gene expression in a family of flies (Diptera: Stratiomyidae), that inhabits extreme environments (including the proximity of volcanoes), and found that HSPs are facultatively upregulated in these flies. This constitutes evidence for evolutionary adaptation occurring in a mechanism geared towards short-term stress response. The HSPA14 gene which encodes the Heat Shock 70kDa Protein 14, is one of these facultatively upregulated HSP genes in Stratiomyidae. LEDGF is a regulatory protein with the capability to upregulate general transcription, and specifically the transcription of HSPs under stress (thermal or oxidative, Sharma et al., 2000). It is hypothetically linked to the HSP functional network via the human SUMO gene (the chicken equivalent could not be identified). HSP47/SERPINH1 was found to be the most commonly upregulated gene that trout fish express in gills under heat stress (Rebl et al., 2013).

Candidate markers within the Aquaporin functional network, and their association with water homeostasis

Water channels, cryoprotectants transport. Another group of candidate markers can be found within the evolutionary ancient Aquaporin gene family coding for membrane proteins that facilitate water and glycerol transport. All genes of the AQP gene family (with the exception of AQP7) have been found and characterized in non-thermoregulating Zebrafish (Tingaud-Sequeira et al., 2010). AQPs transport water, glycerol, and small solutes across membranes which both is important for heat adaptation (evapotranspiration, preventing desiccation) and cold adaptation (cryoprotectant transport into tissues). In endothermous animals, sweating is a major physiological mechanism of thermal plasticity. It is the process of the excretion of salt and water through sweat glands in order to cool down the body via evaporative cooling. Evaporative cooling is the process by which air temperature is lowered by adding water vapor. AQP5 and AQP3 have been found to be expressed in sweat glands of rats, where AQP3 was expressed in the basal levels of the epidermis, and AQP5 was expressed in sweat glands (Nesjum et al., 2002). Similar evaporative cooling is one of the few known processes of thermoregulation in ectothermic vertebrates such as amphibians and reptiles (Tattersall, Cadena & Skinner, 2006). Instead of sweat glands, water evaporates here through the mucosa of the mouth, or through the skin. In anuran amphibians, AQP3 is expressed in osmoregulatory organs and has been identified as the site of transepithelial water exit (Suzuki & Tanaka, 2009). In humans, AQP3 has furthermore found to be involved in glycerol transport in the epidermis (Hara-Chikuma & Verkum, 2008). Suzuki & Tanaka (2009) speculated that AQP3 expression in the frog epidermis is related to glycerol transport. A known mechanism of cold adaptation in ectothermic vertebrates is the prevention of blood crystallization by blood supercooling, usually by blood glucose, glycogen, or glycerol deposition (reviewed in Constanzo, 2011). Zimmerman et al. (2007) found that accumulation of glycerol in frogs during cold acclimation, which is secreted by the liver, was related to expression of aquaporin (including AQP3) gene expression. Consequently, the AQP5 gene is a candidate marker to study thermal adaptation—both to heat and to cold—in amphibians. However, so far there is little evidence for cryoprotectants in reptiles (Storey, 2006). Aquaporin-5 (AQP5) is a protein that forms water specific channels. Three genes for AQP5 are present in the genome of Anolis carolinensis. In humans, AQP5 are localized and expressed in the secretory and lachrymal glands, kidney, lungs and nerves. The AQP5 is modified in yaks (Bos mutus), bovines, as adaptation to survival in high altitude (Qiu et al., 2013) that is both characterized by low oxygen as also cold temperatures.

Candidate markers within the ADORA functional network, and their association with oxygen transport

Signal relay (GO), Muscle contraction and relaxation (GO), Vasodilation, vasoconstriction, blood circulation, blood pressure regulation (GO). Adenosine is an agonistic neurotransmitter present in many vertebrate lineages, including lizards (Michaelidis, Loumbourdis & Kapaki, 2002). In one of the few existing studies demonstrating metabolic thermoregulation in ectothermic vertebrates, Petersen, Gleeson & Scholnick (2003) found that if oxygen is scarce (less than 10%), reptiles (Anolis sagrei and Dipsosaurus dorsalis) are able to downregulate their internal body temperature, even in low ambient temperatures. This mechanism has the putative function of protecting the body against ambient hypoxia. After oxygen is available again, the effect will be abolished and body temperature restored to environmental temperature. Administrating an antagonist to the adenosine receptor prevented this alteration of the thermoregulatory set point. Several adenosine receptors exist in vertebrates that are candidate genes to studying this thermoregulatory behavior, encoded by the genes ADORA1, 2A, 2B, 3 (humans and chicken). In Anolis carolinensis, ADORA1 and ADORA2 seem to be duplicated (Table S1) while ADORA2B is not annotated yet, and ADORA3 is potentially absent. Functional Network clustering in the chicken found the ADORA2A and ADORA2B to be functionally linked to ADRB2 via another signal transducer, GNAL. ADRB2 is another candidate marker for thermal adaptation or plasticity: the encoded Beta-2 adrenergic receptor is involved in nonshivering thermogenesis in primates (Takenaka et al., 2012), and plays a role in human asthma and obesity (Table S1). In humans, both ADRB2 and ADRB3 are involved in lipolysis and thermogenesis and cause differences in energy expenditure (Girardier & Seydoux, 1981; Takenaka et al., 2012). Adrenergic-receptor beta3 (ADRB3) is located mainly on the surface of visceral and brown adipose cells and promotes lipolysis and thermogenesis by noradrenaline release from the sympathetic nerves stimulated by cold temperature or food consumption. ADRB3 is also present in the Anolis carolinensis genome. This gene therefore is likely linked to the functional theme of Lipoprotein metabolism.

Conclusions

In this study, we constructed a chicken multimodal network of functional relationships from a list of published markers for thermal adaptation in vertebrates. This network was more organized into functional pathways, more functionally associated, and faster in information exchange than expected by chance. While some of the markers might be exclusively related to thermoregulation via BAT in mammals, thermal adaptation has been shown to act on any other component of this network in other vertebrate lineages. Surprisingly, markers initially identified from diverse lineages of vertebrates such as human and fish were all in close functional relationship with each other. This indicates that the general genetic functional network for thermoregulation and/or thermal adaptation to the environment might be regulated via similar evolutionarily conserved pathways in different vertebrate lineages. We were able to identify seven functions that were statistically overrepresented in this network, corresponding to four of our originally predicted functions plus three unpredicted functions. We describe this network as multimodal: central regulator genes with the function of relaying thermal signal (1) affect genes with different cellular functions, namely (2) lipoprotein metabolism, (3) membrane channels, (4) stress response, (5) response to oxidative stress, (6) muscle contraction and relaxation, and (7) vasodilation, -constiction and regulation of blood pressure. Behavioral thermoregulation and distribution area shifts are expected to be the primary response of vertebrate ectotherms to changes in the thermal environment. In addition, the functional genetic network established herein provides a new resource for the study of adaptive pathways that exist beyond plastic responses to thermal environmental change in vertebrate ectotherms via experimental or comparative genomic studies. Further research should be directed towards verifying thermal adaptation in candidate markers and towards investigating the genetic basis of thermoregulatory behavior, in order to obtain a comprehensive understanding of the functional genetic basis of thermal adaptation in vertebrate ectotherms.

Supplemental Information

Supplemental Information 1 Flow chart of the process of identifying candidate functional pathways that are likely adaptive to changes in the thermal environment across vertebrates. Question marks denote hypothesis testing

Yellow boxes denote research resources provided in this paper.

Click here for additional data file.

Figure S1 Detailed version of network presented in Figure 2

Green - candidate markers, beige - neigbors

Click here for additional data file.

Table S1 List of candidate genes involved in thermal acclimation or adaptation in different vertebrate lineages

Gene descriptions for humans were obtained from RefSeq via genecards.org

Click here for additional data file.

We thank the Bethune-Cookman University Biology and Chemistry faculty members as well as Academic Affairs for their continued efforts in promoting undergraduate research excellence, and for helpful discussions.

Additional Information and Declarations

Competing Interests

Author Contributions

Data Deposition

The authors declare there are no competing interests.

Katharina C. Wollenberg Valero conceived and designed the experiments, performed the experiments, analyzed the data, wrote the paper, prepared figures and/or tables.

Rachana Pathak, Indira Prajapati, Shannon Bankston, Aprylle Thompson and Jaytriece Usher performed the experiments, analyzed the data, reviewed drafts of the paper.

Raphael D. Isokpehi conceived and designed the experiments, performed the experiments, analyzed the data, contributed reagents/materials/analysis tools, reviewed drafts of the paper.

The following information was supplied regarding the deposition of related data:

An online resource has been created, accessible under: https://public.tableausoftware.com/views/thermal_adapt/gene_list.

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
