# Peer review of "A candidate multimodal functional genetic network for thermal adaptation"

_PeerJ, doi:10.7717/peerj.578_

## Round 0.1 · original submission · Major Revisions

We have now received two reviews and I have read the paper carefully myself. Both reviewers recommend ‘major revision’ and I agree. To begin with I found the rationale very difficult to follow. As far as I could tell the idea is to identify candidate genes in humans based on broad areas in which thermoregulatory compensation may occur (which appears to include elements such as shivering that I don’t think lizards do). These are then used to reconstruct likely functional pathways. This information is then used to find homologues in the chicken and lizard and to see whether certain GO codes are over-represented. The (huge) problem is that there does not seem to be any test for fit to the hypothesis. For example, a gene involved generally in stress response may well also have close functional links to genes involved in response to oxidative stress. Such genes may well be conserved between human, lizard and chicken but there is nothing in this reasoning that ensures involvement in thermoregulation or adaptive response to thermal challenge beyond the original broad choice. As one Referee points out, the rationale therefore appears circular.

To me it is vital to include one or more independent tests for significant association with thermal habitat. One possibility is to repeat the analysis using similar classes of gene NOT thought to be involved in relaying thermal signals and then to test the hypothesis that such genes are in much weaker functional relationships than those chosen as candidate thermal genes. Another test might be to look to see if the rates of evolution of adjacent genes in the functional network are correlated. Or one could look for signatures of accelerated evolution of key genes in species that have experienced a large change in thermal ecology. However it is done, this paper only becomes valuable (in the current context publishable) if there is a clear demonstration (a) that functional links between the genes chosen are significantly stronger than might be expected by chance and (b) that there is a link to thermal adaptation that is shown independently of the original choices.

The Reviewers both make a number of other useful suggestions and comments which should help improve a revised version. I would add to these a need for greater clarity in the rationale: why use chicken and lizard but not, say, add in a frog and a fish? What does the comparison between the three species actually tell you?

·

Basic reporting

"No Comments"

Experimental design

"No Comments"

Validity of the findings

"No Comments"

Additional comments

This manuscript integrates information from various sources to detect candidate genes for thermal adaptation. The authors define a number of functional categories that likely cover most candidate genes, and use the chicken genome as a model. This makes sense because many elements of avian thermoregulatory mechanisms are present in reptiles. The authors retrieved a number of genetically overrepresented functions in Gene Ontology associated with lipoprotein metabolism, membrane channels, stress response, etc.

The work is novel and interesting. I only have one concern, which I think is important. Behavioral plasticity of habitat use is a key ingredient for thermoregulation in ectotherms, and it is obvious that this component is lost in the present manuscript because of the design of the study. The importance of thermoregulatory behaviors has been recently highlighted in the paper “Sunday et al. 2014. Thermal-safety margins and the necessity of thermoregulatory behavior across latitude and elevation. PNAS 111: 5610–5615.” These authors conclude that to survive climate warming ectotherms in most areas may need to rely on behavioral thermoregulation. Therefore, the present authors have included a list of candidate genes for thermal adaptation in table 1 BUT these genes may be relatively irrelevant to the way ectotherms can cope with high temperatures. Perhaps the most relevant genes in this case are those associated to the detection of temperature to avoid it if it is low or high by moving to another area (if possible).

So although I think the manuscript is interesting and potentially useful, I would encourage the authors to add a paragraph highlighting the limitations of the study. This is particularly relevant in view of the role that behavioral thermoregulation plays in ectotherms to cope with rising temperatures.

Miscellaneous comments:

p.2 The introducing paragraph “The process of thermal adaptation has, in the recent climate change debate, fueled many studies in evolutionary biology that study thermal adaptation in animals” is redundant. It needs some rewording as (e.g.) “The human-induced climate change has recently fueled a renewal interest in the study of thermal adaptation ….”.

p.2 “Temperature affects animals primarily via its effects on biochemical reaction rates (Tattersall et al., 2012)”. A more appropriate reference would be the classic book “Hochachka, P.W. and Somero, G.N. (2002) Biochemical Adaptation. Mechanism and Process in Physiological Evolution. Oxford University Press, New York.”

p.2 “Ghalambor et al. 2006. Are mountain passes higher in the tropics? Janzen's hypothesis revisited. Integrative and Comparative Biology 46:5-17” provides an up to date revision of Janzen’s paper “Why Mountain Passes Are Higher in the Tropics.”

p.3 Note: “acclimation” is usually used to describe the changes within an organism in response to experimentally induced changes in particular climatic factors such as ambient temperature. The term “acclimatization” is used when these changes within the organism occur in response to changes in the natural climate. In the context of the paper I think the term acclimatization is more appropriate. However, the distinction the authors make between acclimation (acclimatization) and plasticity may seem arbitrary as the term phenotypic plasticity is broadly used to describe all phenotypic responses to environmental change (also including acclimation or acclimatization). To sum up, I think it is more appropriate to only distinguish between plasticity and evolution (genetic adaptation) as organismal responses to changes in the environment.

p.3-4 The paper “Vedder O, Bouwhuis S, Sheldon BC (2013) Quantitative Assessment of the Importance of Phenotypic Plasticity in Adaptation to Climate Change in Wild Bird Populations. PLoS Biol 11(7): e1001605” should be cited as providing evidence that evolution offers the chance for short-lived birds to adapt at the rate of climate change that is expected over the next century.

p.4 “Leppäluoto et al., 2005” is missing in the references.

p.5 “It is not known which mechanisms ectotherms use to maintain body heat except for behavioral THERMOREGULATION.”

Reviewer 2 ·

Basic reporting

The authors introduced in an appropriate way the theme of cold adaptation through thermal regulation in endothermic and ectothermic animals, which is definitely of primary importance to understand the overall mechanisms underlying thermal adaptation in animals. Furthermore they aim at finding patterns of adaptation shared by distantly related species, hence stressing the need for a thorough understanding of the baseline mechanisms of such adaptation. However there are several major issues that the author should address in order to build up convincing evidences of shared mechanisms of adaptation between the examined species (namely: Homo sapiens, Gallus gallus and Anolis carolinensis). Such issues are going to be described in details in the following sections of this review.
While the overall format of the manuscript and the English are satisfactory, the lists of candidate genes and their putative implication in thermal regulations provided in pages 10-20 is definitely too long and should be either moved to the supplementary materials or appropriately expanded in a review paper on the subject.
Concerning the graphical representation of their findings, Figure 1 (the phylogenetic tree linking the 3 analysed species) could be substantially improved by providing details on the way it was calculated and, if appropriate, actual measures of phylogenetic distances. Indeed at present the represented tree is not different from a best guess sketch of a three given a baseline morphological knowledge of the evolutionary relationships between the three species.
Figure 2 (and its derived sub-figures 3 and 4 as well as supplementary figure 1) could benefit from a clarification on the nature of the white/yellow labelled proteins. If indeed, as stated by the authors towards the end of page 7, all the while proteins “were added to the algorithm-generated networks in CorelDraw”, then the proposed protein-protein interactions would make sense only in humans. When removing the white proteins from these figures, indeed only a small proportion of the candidate markers display actual functional interactions between each other.

Experimental design

The authors propose to retrieve the broadest possible list of candidate genes for thermal regulation through a literature search of all the vertebrate species. They also define a list of five potentially relevant functional categories which might play a role in thermal regulation. Of these, three were found over-represented in the identified list of candidate markers together with three additional functional categories. This finding would be relevant to characterize the thermal regulation pathway, however the authors do not rule out potential circularities in their arguments. If indeed the literature search for the candidate genes was performed either explicitly using the 5 functional categories (or parts) as keywords or simply looking for keywords related to these 5 categories, the enrichment signal is nothing but a consequence of this circularity. Therefore a clear explanation of the literature search procedure is necessary to rule out this potential source of artefact in the author’s conclusion.

Another potential source of artefacts in the experimental design in the deliberate insertion of genes only found in the human functional networks in networks based on the chicken protein-protein interaction database. This operation can bias the authors conclusions in two ways: firstly it can introduce spurious functional links which may artificially increase the interconnections between candidate markers; secondly it may lead to the wrong conclusions that these markers are functionally connected in both chicken and humans (hence underlying a common pattern of thermal regulation), while actually most of the connections observed in chicken were simply borrowed from the human network.

Validity of the findings

In the present manuscript the authors propose three main findings:
- a comprehensive list of genes potentially implicated with thermal regulation in at least one vertebrate species. The presence of many of these genes as homologous in three distantly related species (lizard, chicken and human) is interpreted by the authors as a plausible evidence of a conserved, broad mechanism of thermal regulation between these three species and potentially across most vertebrates.

- While the presence of homologs with a similar function in the three surveyed species is indeed a good evidence of a conserved machinery, no conclusion can be drawn on the conservation of functional connections between markers if, as described in the previous comments, most of the functional connections are only presents in humans (white proteins in figure 2 and onwards)

- The authors also flag the enrichment of the candidate genes for six functional categories of which three present in the category list putatively used for the gene search. Unless the authors rule out putative circularities between the literature search process and the finding of GO enrichment, at least 3 out of the 6 enriched categories should not be used in further interpretation of their results.

Additional comments

No additional comments

---

## Round 0.2 · Minor Revisions

I have now read your rebuttal. This is (obviously!) not my area, but I now understand the paper much better. My feeling is that this remains a highly technical paper and that, even now, though a lot clearer, it is still not trivial for non-specialists to understand exactly what is being done. I have one further suggestion that you might like to consider, namely to include a cartoon flow diagram to illustrate the process and to make the hypothesis being tested crystal clear. You might feel this is unnecessary, but I do think it would help make your work approachable to a much larger audience.

The randomization test seems to have worked well and makes me much more convinced the hypothesis is supported. There are two slight tweaks. First, I would like to see a statement concerning exclusion / inclusion of genes with unknown function. I presume pseudogenes / unknown function genes / micro RNAs etc. were excluded, otherwise it is no surprise that fewer connections were made. A statement about the criteria used should be added. If no exclusions were made, then the analysis needs to be repeated using a fair comparison set. Ideally this would involve matching for connectivity, since it is possible that thermoregulatory genes are generally better categorised and as a result have more connections compared with ‘other genes’. At the very least, genes which might be thought a priori to have low connectivity should be excluded. Finally, your new figure is nice but would benefit from exact P-values, which are surely easy to generate.

---

## Round 0.3 · accepted · Accept

Some of the figures look a bit pallid / unclear on my computer, they may need tightening for final publication. Apart from that, great.